Applied and Environmental Science

mSystems®

# Elucidation of Regulatory Modes for Five Two-Component Systems in *Escherichia coli* Reveals Novel Relationships

Kumari Sonal Choudhary,[a] Julia A. Kleinmanns,[a] Katherine Decker,[a] Anand V. Sastry,[a] Ye Gao,[a] Richard Szubin,[a] Yara Seif,[a] Bernhard O. Palsson[a,b]

[a]Department of Bioengineering, University of California, San Diego, San Diego, California, USA
[b]Novo Nordisk Foundation Center for Biosustainability, Technical University of Denmark, Lyngby, Denmark

Kumari Sonal Choudhary, Julia A. Kleinmanns, and Katherine Decker contributed equally to this work. Author order was determined based on contribution to the conceptualization, implementation, investigation, and writing of the manuscript.

**ABSTRACT** *Escherichia coli* uses two-component systems (TCSs) to respond to environmental signals. TCSs affect gene expression and are parts of *E. coli*'s global transcriptional regulatory network (TRN). Here, we identified the regulons of five TCSs in *E. coli* MG1655: BaeSR and CpxAR, which were stimulated by ethanol stress; KdpDE and PhoRB, induced by limiting potassium and phosphate, respectively; and ZraSR, stimulated by zinc. We analyzed RNA-seq data using independent component analysis (ICA). ChIP-exo data were used to validate condition-specific target gene binding sites. Based on these data, we do the following: (i) identify the target genes for each TCS; (ii) show how the target genes are transcribed in response to stimulus; and (iii) reveal novel relationships between TCSs, which indicate noncognate inducers for various response regulators, such as BaeR to iron starvation, CpxR to phosphate limitation, and PhoB and ZraR to cell envelope stress. Our understanding of the TRN in *E. coli* is thus notably expanded.

**IMPORTANCE** *E. coli* is a common commensal microbe found in the human gut microenvironment; however, some strains cause diseases like diarrhea, urinary tract infections, and meningitis. *E. coli*'s two-component systems (TCSs) modulate target gene expression, especially related to virulence, pathogenesis, and antimicrobial peptides, in response to environmental stimuli. Thus, it is of utmost importance to understand the transcriptional regulation of TCSs to infer bacterial environmental adaptation and disease pathogenicity. Utilizing a combinatorial approach integrating RNA sequencing (RNA-seq), independent component analysis, chromatin immunoprecipitation coupled with exonuclease treatment (ChIP-exo), and data mining, we suggest five different modes of TCS transcriptional regulation. Our data further highlight noncognate inducers of TCSs, which emphasizes the cross-regulatory nature of TCSs in *E. coli* and suggests that TCSs may have a role beyond their cognate functionalities. In summary, these results can lead to an understanding of the metabolic capabilities of bacteria and correctly predict complex phenotype under diverse conditions, especially when further incorporated with genome-scale metabolic models.

**KEYWORDS** two-component systems, *E. coli*, independent component analysis, transcriptomics, ChIP-exo, transcriptional regulatory network, gene targets

Address correspondence to Bernhard O. Palsson, palsson@ucsd.edu.

Integrative systems biology approach identifies different modes of transcriptional regulation and novel inducers of two-component systems in E. coli.

**B**acterial survival and resilience across diverse conditions rely upon environmental sensing and a corresponding response. One pervasive biological design toward this goal consists of a histidine kinase unit to sense the environment and a related response regulator unit to receive the signal and translate it into gene expression changes. This signaling process is known as a two-component system (TCS) (1). In the case of *Escherichia coli* strain K-12 MG1655, there are 30 histidine kinases and 32 response

regulators involved in 29 complete two-component systems that mediate responses to various environmental stimuli such as metal sensing, cell envelope stress, acid stress, and pH stress (2). The cell envelope is an important barrier between bacteria and their surrounding environment and is exposed to a variety of stresses and stimuli. In *E. coli* MG1655, BaeSR and CpxAR are two TCSs that are each instrumental in regulating the response to envelope stress (3). KdpDE and ZraSR are key regulators in maintaining homeostasis for potassium and zinc, respectively, and PhoRB has a role in bacterial pathogenesis and phosphate homeostasis (4). Understanding these elements is important for developing an overarching knowledge of bacterial regulatory networks. A detailed reconstruction of the transcriptional regulatory network (TRN) contribution of these systems in response to a specific stimulus will help us discover different modes of transcriptional regulation. Furthermore, this can unravel cross-regulation and complex relationships among TCS systems.

Multiple tools exist to investigate the regulons and activities of transcription factors and TCSs. In this study, we focused on two data types: chromatin immunoprecipitation coupled with exonuclease treatment (ChIP-exo) binding peaks and gene expression profiles from RNA sequencing (RNA-seq). ChIP-exo identifies genome-wide binding locations for response regulators with high precision (5). Although ChIP methods provide direct binding evidence for the interactions between transcription factors (TFs) and the genome, they also tend to identify off-target, nonspecific binding events that may not affect downstream gene expression (6). In order to reduce the false-positive rate from ChIP-exo, gene expression profiles can identify differentially expressed genes (DEGs) in response to appropriate stimuli or response regulator knockouts. However, some DEGs may be affected by indirect regulation. To circumvent this issue, we also used independent component analysis (ICA), an unsupervised machine learning algorithm that decomposes gene expression data, to find independently modulated sets of genes, or iModulons (Fig. 1A), that closely resemble known regulons (7).

A previous application of ICA to PRECISE (a high-quality RNA-seq gene expression compendium) (7) identified nine iModulons that corresponded to 11 TCS regulons. We noticed that two sets of TCS pairs were grouped together: YpdAB and BtsRS grouped together, and HprSR and CusSR grouped together (7). Interestingly, YpdAB and BtsRS respond to the same inducer (pyruvate). Conversely, HprSR and CusSR both regulate the same set of genes but have different inducing signals. This second observation is in agreement with previous studies (8) and is indicative of cross-regulation between the two TCSs wherein they have the same regulon. ICA generates iModulon activity levels to determine gene expression changes relative to a specific condition. Indeed, these previous analyses showed that ICA can be aptly used to describe condition-specific activities (YpdAB and BtsRS) and provide a detailed understanding of tightly cross-regulated TRNs (HprSR and CusSR).

In this study, we aimed to characterize global TRN contributions of five TCSs in *E. coli* MG1655: BaeSR, CpxAR, KdpDE, PhoRB, and ZraSR. We used three methods (ICA, ChIP-exo, and differential expression of genes [DEGs]) to reveal which genes were regulated by the TCS and characterized the modes of transcriptional regulation into five categories (Fig. 1B): (i) functional direct binding, (ii) indirect targets, (iii) nonfunctional/spurious binding, (iv) cross-regulated genes, and (v) hypothetically functional binding. Finally, we used the iModulon activities to characterize TCS activation across hundreds of experimental conditions to identify interconnectedness among TCSs.

## RESULTS

We generated knockout mutants for each of the five two-component systems (BaeSR, CpxAR, KdpDE, PhoRB, and ZraSR) in *E. coli* K-12 strain MG1655 (Table 1) and collected RNA-seq data for both wild-type (WT) and knockout mutants under stimulated and unstimulated conditions, using specific stimuli for each TCS. We combined these expression profiles with 278 additional expression profiles previously generated (called PRECISE [7]), and applied the ICA algorithm to the combined data set (see Materials and Methods) to identify 101 iModulons. Of these 101 iModulons, 90 iModu-

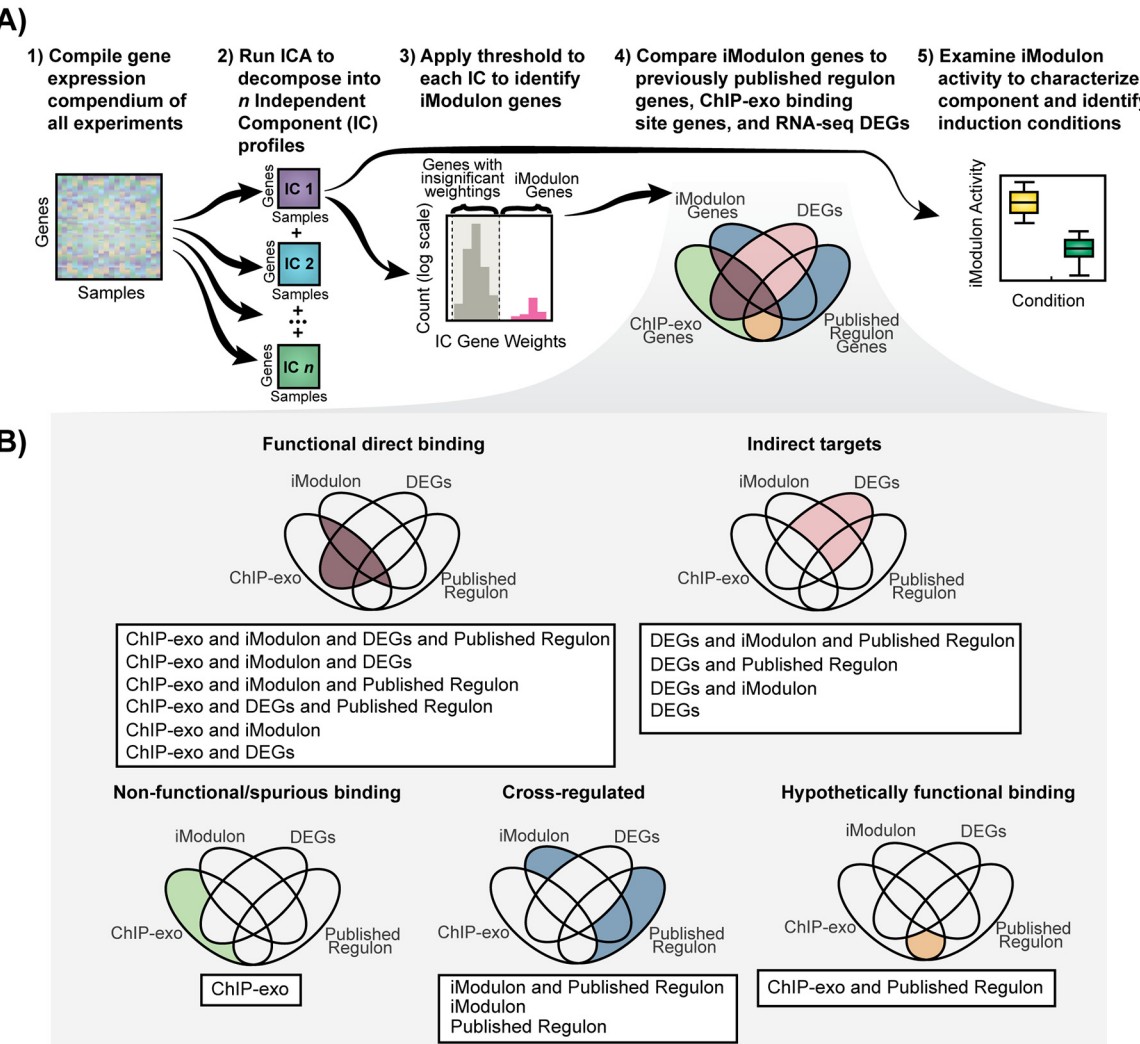

**FIG 1** (A) Workflow used for each gene expression data set. (B) Categories of regulation by two-component systems in *E. coli*.

lons mapped to the previous set of 92 iModulons from PRECISE (7). Eleven iModulons were specifically activated by the inclusion of the TCS-related expression profiles, four of which represented TCSs (BaeSR, KdpDE, PhoRB, and ZraSR). The CpxR iModulon was among the 90 iModulons previously detected in PRECISE and was retained upon addition of the new expression profiles. In total, 15 of the 101 iModulons represented TCSs (see Data Set S1 in the supplemental material). For further analysis, we focused on the five target TCSs (BaeSR, CpxAR, KdpDE, PhoRB, and ZraSR).

To identify the true regulons for each TCS, we compared four sets of genes: (i) genes in the iModulon associated with the respective TCS, (ii) genes with ChIP-exo deter-mined upstream binding sites, (iii) genes that are differentially expressed between wild-type and knockout mutants under stimulated conditions, and (iv) genes in previ-ously published regulons in RegulonDB (9) (Fig. 1B). Each method alone is prone to false-positive results and biases, but together they provide multidimensional informa-tion that can be probed to identify a complete, high-confidence regulon (Fig. 1B). For example, differentially expressed genes (DEGs) without direct ChIP-exo evidence are likely targets of downstream, indirect regulation wherein one TF may regulate a set of genes which in turn may lead to regulation of a second set of genes. On the other hand, ChIP-exo binding sites upstream of genes that are absent from both the iModulon and the DEG set are likely nonfunctional. Such integrated analysis led us to describe the regulatory network of each selected two-component system and place it in five

**TABLE 1** List of strains and culture conditions used in this study

| Strain | Phenotype | Stimulation condition | Reference | Unstimulated condition | Reference |
|---|---|---|---|---|---|
| E. coli K-12 MG1655 | Wild type | (i) LB medium + 5% ethanol | This study | (i) LB medium | This study |
| | | (ii) Tris-maleic acid minimal medium (TMA) + 0.1 mM KCl | | (ii) Tris-maleic acid minimal medium (TMA) + 115 mM KCl | |
| | | (iii) M9 minimal medium without phosphate (M9-P) | | (iii) M9 minimal medium | |
| | | (iv) LB medium + 1 mM ZnCl$_2$ | | | |
| ΔbaeR | baeR knockout mutant of E. coli K-12 MG1655 | LB medium + 5% ethanol | This study | LB medium | This study |
| ΔcpxR | cpxR knockout mutant of E. coli K-12 MG1655 | LB medium + 5% ethanol | This study | LB medium | This study |
| ΔkdpE | kdpE knockout mutant of E. coli K-12 MG1655 | Tris-maleic acid minimal medium (TMA) + 0.1 mM KCl | This study | Tris-maleic acid minimal medium (TMA) + 115 mM KCl | This study |
| ΔphoB | phoB knockout mutant of E. coli K-12 MG1655 | M9 minimal medium without phosphate (M9-P) | This study | M9 minimal medium | 7 |
| ΔzraR | zraR knockout mutant of E. coli K-12 MG1655 | LB medium + 1 mM ZnCl$_2$ | This study | LB medium | This study |

categories (Fig. 1B): (i) functional direct binding (genes that were observed in either iModulons, published regulons, or DEGs, and a ChIP-exo binding peak was observed upstream), (ii) indirect targets (genes that were differentially expressed under each specific condition, but no binding peaks were observed upstream), (iii) nonfunctional/ spurious binding (peaks upstream of genes that were neither differentially expressed nor had iModulons and/or published regulons associated with them [Such peaks may have arisen due to off-target recognition or due to hot spots that tend to be recognized by many TFs.]), (iv) cross-regulated genes (genes that were observed in iModulons and/or published regulons but were not part of functional direct targets and indirect targets), and (v) hypothetically functional binding (genes that were found in the published regulon and showed ChIP-exo binding peaks but were not differentially expressed). These genes may be a part of a direct regulatory network, but the data were not supported by transcriptomics; therefore, we categorized them into hypothetical functional binding.

**Reconstruction of CpxR and BaeR regulatory responses under ethanol stress in E. coli.** BaeR and CpxR are activated by a wide range of stressors (10–14) and play a regulatory role in the envelope stress response (ESR) system (10, 15–17). We stimulated the wild-type, ΔbaeR, and ΔcpxR strains with 5% ethanol stress, which has been identified as a particularly effective inducer (18) to elucidate their regulons.

**(i) BaeR.** BaeR is involved in multidrug resistance through regulation of the MdtABC efflux pump (19–21). However, BaeR has also been shown to impact genes related to many other functions, such as signal transduction, chemotactic responses, flagellar biosynthesis, and maltose transport (22). Despite this range of functionality, there are only eight genes in the previously published BaeR regulon. The BaeR iModulon exhibited 37.5% (3 out of 8) overlap with these published regulon genes, consisting of spy, baeR, and mdtA genes (Table 2). Five of the published regulon genes (mdtB, mdtC, mdtD/iceT, baeS, and acrD) were not identified in the BaeR iModulon. This discrepancy was most likely due to differences in BaeR induction conditions across the published results (10–13) or the possibility that the three iModulon genes are much more significantly regulated by BaeR than the other five genes in the published regulon. To test this claim, ChIP-exo results were analyzed (Data Set S2) and then compared to the iModulon. ChIP-exo results indicated 21 BaeR binding peaks upstream of 17 operons consisting of 32 genes. We identified binding sites upstream of spy and mdtA genes (mdtABCD-baeSR operon). Since a peak was identified upstream of mdtA gene, which is part of the mdtABCD-baeSR operon (see Materials and Methods), we considered all the genes in that operon to be under the direct regulatory network of BaeR. In total, we

**TABLE 2** List of genes in each iModulon and published regulon and their direct targets

| Response regulator | iModulon genes[a] | Targets from RegulonDB[a] | Direct targets |
|---|---|---|---|
| BaeR | **baeR**, **mdtA**, **spy** | acrD, **baeR**, baeS, mdtD, **mdtA**, mdtB, mdtC, **spy** | baeR, baeS, mdtD, mdtA, mdtB, mdtC, spy, intF, tnaA |
| CpxR | alx, **baeR**, **cpxP**, **cpxR**, **dgcZ**, **ftnB**, **ldtC**, mdtJ, raiA, **tomB**, yagU, **yccA**, **yebE**, yjfN, yncJ, yobB | acrD, aroG, bacA, **baeR**, baeS, bamE, cheA, cheW, cpxA, **cpxP**, **cpxR**, csgA, csgB, csgC, csgD, csgE, csgF, csgG, degP, **dgcZ**, dsbA, dsbC, efeU_2, fabZ, **ftnB**, hha, iceT, **ldtC**, ldtD, lpxA, lpxD, marA, marB, marR, mdtA, mdtB, mdtC, motA, motB, mscM, mzrA, ompC, ompF, ppiA, ppiD, psd, rpoE, rpoH, rseA, rseB, rseC, sbmA, skp, slt, spy, srkA, **tomB**, tsr, ung, yaiW, **yccA**, **yebE**, ygaU, yidQ, yqaE, yqjA | cpxA, cpxP, cpxR, yagU, yccA, fimA, fimI, lldD, lldR, ppiA, raiA, ycel |
| KdpE | **kdpA**, **kdpB**, **kdpC**, kdpD, kdpE | **kdpA**, **kdpB**, **kdpC**, kdpF | kdpA, kdpB, kdpC, adeP, uxaA, uxaC |
| PhoB | **phoB**, **phoR**, **pstA**, **pstB**, **pstC**, **pstS** | adiC, amn, argP, asr, cra, cusA, cusB, cusC, cusF, cusR, cusS, eda, feaR, gadW, gadX, hiuH, mipA, ompF, phnC, phnD, phnE_1, phnE_2, phnF, phnG, phnH, phnI, phnJ, phnK, phnL, phnM, phnN, phnO, phnP, phoA, **phoB**, phoE, phoH, phoQ, **phoR**, phoU, pitB, prpR, psiE, psiF, **pstA**, **pstB**, **pstC**, **pstS**, rspR, sbcC, sbcD, tktB, ugpA, ugpB, ugpC, ugpE, ugpQ, waaH, yegH, yhjC | phoB, phoR, pstA, pstB, pstC, pstS, phoU |
| ZraR | **zraP**, **zraR** | **zraP**, **zraR**, zraS | zraP, zraR, zraS, mgtA |

[a]Boldface genes in the "iModulon genes" and "Targets from RegulonDB" columns are shared genes.

designated six genes as functional direct targets of BaeR. No binding peaks were identified upstream of *acrD*, indicating that it is indirectly regulated by BaeR.

Differentially expressed genes were analyzed to further expand the global regulatory network of BaeR. A total of 328 genes were found to be differentially expressed due to the *baeR* knockout under 5% ethanol stress condition (Data Set S3). ChIP-exo peaks were detected upstream of nine of these genes, expanding the number of direct targets from seven to nine. Of these nine targets, two genes, *intF* and *tnaA*, have not been previously identified as a part of the BaeR regulon. However, *tnaA* has previously been shown to contribute to BaeR activity under indole stress (23), as it is involved in metabolizing tryptophan to indole. The remaining 319 differentially expressed genes could either be indirect targets of BaeR (Fig. 1 and Data Set S4) or other transcription factors responsive to ethanol such as CpxR. Functional analysis of indirect targets through COG (cluster of orthologous group) categories confirmed BaeR's manifold activity (Fig. 2).

Overall, 2.74% of DEGs were under the direct regulatory network of BaeR. The remaining 23 genes found through ChIP-exo results could be classified as nonfunctional targets or cases of spurious binding (Fig. 1 and Data Set S4). Binding of these genes may have not led to expression if the induction period or strength was insufficient; alternatively, these genes might have shown ChIP-exo peaks as a result of off-target binding.

**(ii) CpxR.** CpxR is among the most extensively studied of the response regulators, which is unsurprising when considering its host of functions. CpxR has been shown to have a regulatory role in the envelope stress response system (10, 15–17), protein folding and degradation (24–27), pilus assembly and expression (28–30), secretion (15), motility and chemotaxis (15, 31), biofilm development (14, 32), adherence (33, 34), multidrug resistance and efflux (12, 35), porins (36), and copper response (37, 38), among others. Two iModulons were linked to the CpxR regulon: CpxR and CpxR-KO (KO stands for knockout). The CpxR iModulon consists of genes that were regulated specifically under wild-type conditions, and CpxR_KO consists of genes differentially expressed due to CpxR deletion. Collectively, both CpxR iModulons consist of 16 genes that cover diverse functions such as motility, inorganic ion transport and metabolism, and carbohydrate metabolism, among others. Nine of the 16 iModulon genes were previously published regulon genes, which is only a fraction (14%) of CpxR's previously identified 66 genes (9) (Table 2). This provides extra incentive to eventually validate the CpxR iModulon with different induction conditions. Interestingly, the CpxR iModulon

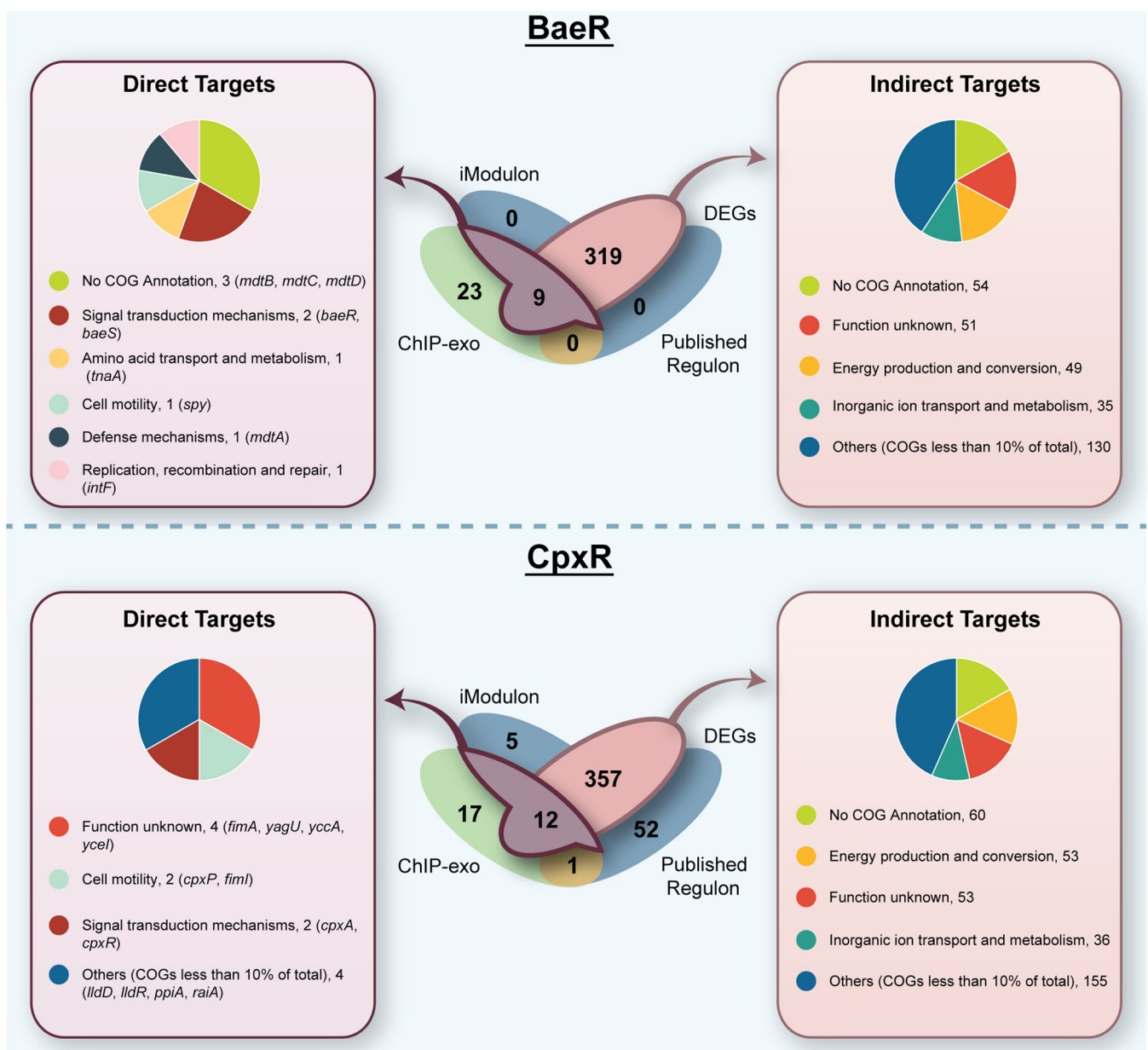

**FIG 2** Comparison of regulatory network in BaeR and CpxR and functional characterization of direct and indirect targets.

consisted of seven new genes that were not previously identified to be part of the CpxR regulon (Table 2). These seven new genes may indicate expansion of the CpxR regulon specific to ethanol stress. To validate this hypothesis, ChIP-exo peaks were examined. We were able to identify 44 binding peaks upstream of 16 operons (consisting of 30 genes). Of these genes, five were in the CpxR iModulon (*cpxP*, *cpxR*, *raiA*, *yagU*, and *yccA*). Two genes (*raiA* and *yagU*) in the CpxR iModulon with associated binding peaks had not been previously identified as part of the CpxR regulon. The five overlapping ChIP-exo/iModulon/regulon genes were classified as direct targets of CpxR under ethanol. For the remaining five iModulon genes that are not previously published regulon genes and do not have binding peaks identified, 60% are part of the uncharacterized "y-ome" (39). These genes may be the result of cross-regulation between different pathways responding to ethanol and put under the category of cross-regulatory network.

We inspected DEGs to further understand the nature of the five new iModulon genes and to expand the global regulatory network of CpxR. Differential gene expres-

sion analysis expanded the indirect targets of CpxR to include 368 genes. Comparison of ChIP-exo binding peaks and differentially expressed genes of the *cpxR* knockout versus the wild type under 5% ethanol stress condition revealed seven additional genes (*cpxA*, *fimA*, *fimI*, *lldD*, *lldR*, *ppiA*, and *ycel*) that fall under the direct regulatory network of CpxR, expanding this network to 12 genes. The remaining 357 DEGs formed the indirect regulon of CpxR, including eight genes from the previously published regulon (Fig. 2 and Data Set S4). A diverse group of COG categories corroborated CpxR's regulatory role.

Interestingly, among the 368 differentially expressed genes, 194 were differentially expressed in both knockout strains (*baeR* and *cpxR*), suggesting nonspecific targets of BaeR and CpxR and high cross-regulation between these two pathways. The remaining 17 ChIP-exo binding peaks overlapped with one previously published regulon gene (*ompC*) which was not part of either the DEGs or iModulons. This suggested that *ompC* was neither directly nor indirectly affected by *cpxR* knockout under ethanol stress, and therefore is considered to be part of the "hypothetically functional binding" network (Data Set S4). Five iModulon genes (*alx*, *mdtJ*, *yjfN*, *ylil*, and *yobB*) did not form part of direct and indirect targets but instead formed part of the network that is involved in cross-regulation. The remaining 52 previously published regulon genes that did not have binding sites upstream and were not differentially expressed also formed part of a cross-regulatory network (Data Set S4).

**Reconstruction of KdpE, PhoB, and ZraR transcriptional regulatory network in *E. coli*.** Decoding the role of metal and nutrient sensors in the transcriptional regulatory network may provide a deeper understanding of how bacteria utilize these systems to sense nutrients and ionic strength of the environment. These mechanisms play a crucial role in the organism's ability to adapt to various environmental niches. The KdpE, PhoB, and ZraR two-component systems were selected for further analysis in this category.

**(i) KdpE.** The KdpE response regulator's sole target is the *kdpFABC* operon, which has been shown to activate the potassium uptake system during low potassium conditions (40, 41) or salt stress (42, 43). To induce KdpE, a small amount of potassium (0.1 mM KCl) was added to the cells in Tris-maleic acid minimal medium (TMA), which aligns with previous experiments where successful induction was achieved (44, 45). The KdpE iModulon included the expected *kdpA*, *kdpB*, and *kdpC* potassium uptake genes; however, the iModulon does not include the fourth gene in the *kdpFABC* operon, *kdpF* (Table 2). This result provides validation of previous studies that have instead described the target operon as *kdpABC* (43, 46). *kdpF* may function only as a stabilizer of the transporter complex (47), and the gene may also just be a less vital component of the regulon. Apart from the *kdpABC* operon, the only other KdpE iModulon genes include those that code for the TCS itself, *kdpD* and *kdpE*. The small KdpE iModulon confirms the existing knowledge of the response regulator as targeting a single locus in the bacterial genome. Despite the small iModulon size, there were still 31 ChIP-exo binding peaks upstream of 37 genes, including *kdpABC* genes from KdpE iModulon observed. Surprisingly, no binding peaks were observed upstream of two iModulon genes, *kdpDE*; however, *kdpDE* is downstream of the *kdpFABC* operon, which may indicate that *kdpDE* is in the same transcription unit as *kdpABC*. Consistent with this, when comparing 197 DEGs in response to *kdpE* knockout under potassium limited condition (0.1 mM KCl), we did find *kdpDE* to be differentially expressed. Comparing ChIP-exo and DEG results further added three extra genes (*adeP*, *uxaA*, and *uxaC*) to the direct regulatory network of KdpE, amounting to six genes under direct regulation of KdpE. The remaining 191 DEGs can be interpreted as potassium-dependent genes under indirect KdpE regulation. As expected, the COG category "inorganic ion transport" was overrepresented. In addition, the categories "energy production and conversion" and "metabolism and nucleotide transport and metabolism" were also overrepresented. A wide array of COG categories suggests that KdpDE may indirectly regulate genes involved in metal efflux pump and electron transport systems to maintain homeostasis (Fig. 3).

Notably, there was evidence of *kdpF* being down-expressed in the KdpE knockout (log fold change of −3.22) but due to an insignificant *P* value, this was not regarded as

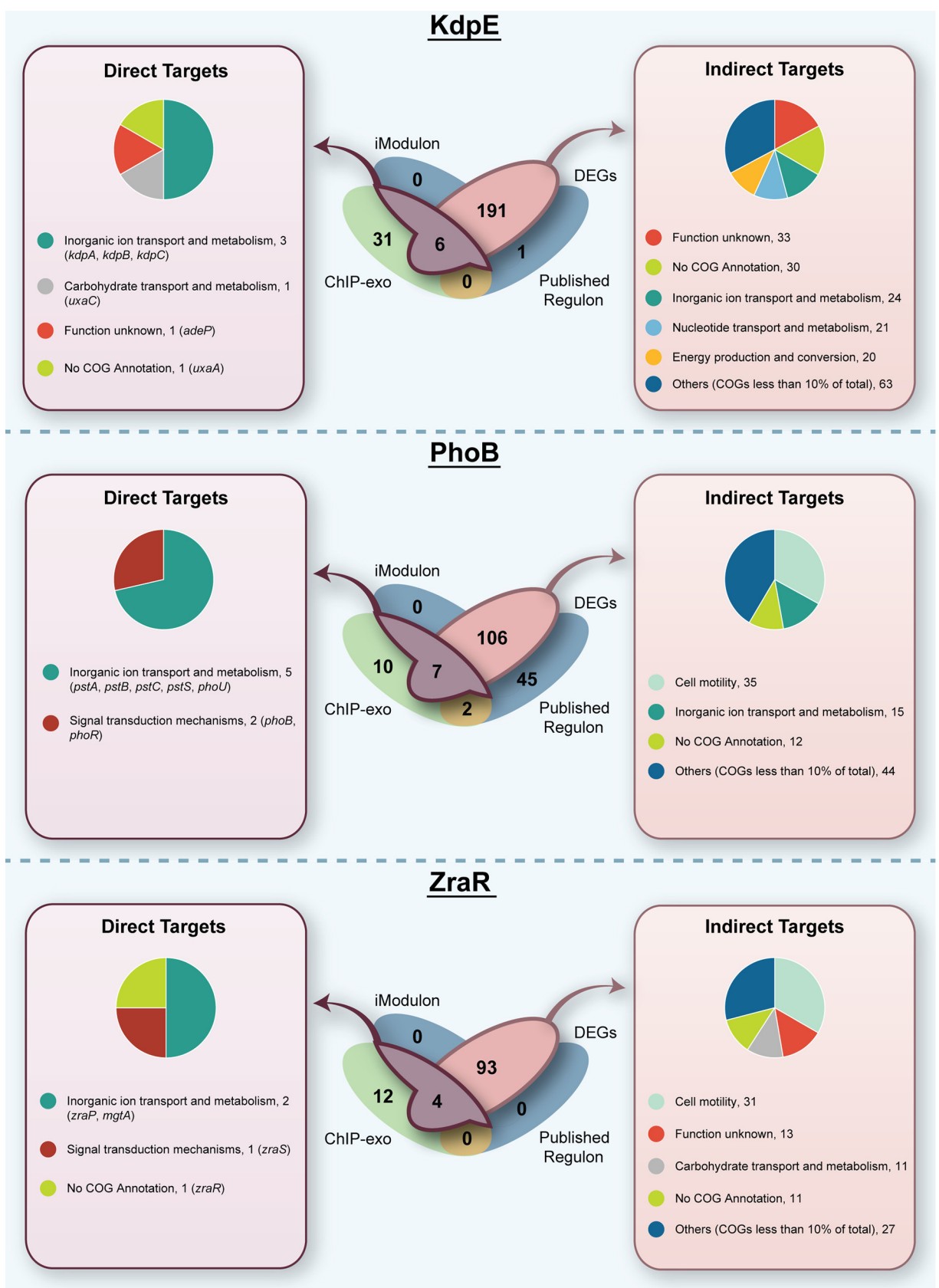

**FIG 3** Comparison of regulatory network in metal sensors and functional characterization of direct and indirect targets.

differentially expressed. Therefore, *kdpF* appears to be the sole gene from the KdpE published regulon that was not identified by DEG analysis in this study (placing it in the category of "cross-regulated" genes). Apart from *kdpF* and genes categorized as direct or indirect targets for KdpDE, there were also 31 genes identified with ChIP-exo that did not overlap with other data sets and were thus categorized as potential spurious binding genes.

**(ii) PhoB.** The PhoB response regulator has been demonstrated to regulate phosphate uptake and metabolism under phosphate-limiting conditions (48–51) and is linked to virulence in pathogenic *E. coli* (4). To induce PhoB via phosphate deficiency, cells were first grown in M9 minimal medium and then washed with and further grown in M9 minimal medium without phosphate (M9-P). The PhoB iModulon consisted of the expected *pstSCAB* operon for regulation of the phosphate uptake system (52–54) and the genes coding for PhoRB itself. In contrast to the modest six-gene iModulon, the previously published PhoB regulon is considerably larger and is thought to consist of 60 genes (55, 56), including all six of the iModulon genes. For the binding site analysis, we found 12 ChIP-exo binding peaks upstream of 17 genes. Six of these genes were included in the iModulon (the *pstSCAB phoBR* operons), indicating direct response to *phoB* knockout. When comparing the 113 DEGs under phosphate-limited condition to the ChIP-exo binding peaks, an additional gene, *phoU* (encoding a phosphate-specific transporter) (which was also part of the previously published regulon) was added to the direct regulatory network of PhoB. The remaining 106 DEGs can be identified as phosphate-dependent genes, including six previously determined regulon genes (*cusB*, *phnC*, *phnD*, *phnN*, *phoA*, and *ugpB*) or indirect targets of PhoB. Interestingly, the COG category "cell motility" was more pronounced than "inorganic ion transport and metabolism," indicative of the involvement of PhoRB in cell migration in addition to adapting to a phosphate-limited environment. Aside from the direct and indirect targets, the remaining 57 genes found to be associated with PhoB included 10 genes that were identified as a potential result of spurious binding, two genes that fell under the hypothetically functional binding category, and 45 genes that were possibly related to the KdpE network via cross-regulation.

**(iii) ZraR.** Finally, we chose to examine ZraR (formerly called HydG), which had been previously characterized as a $\sigma^{54}$-dependent response regulator that activates chaperone ZraP to provide tolerance to high zinc concentrations (57–59). More recently, it has been proposed that ZraR is not directly involved in zinc resistance (60) but rather is simply activated by zinc to achieve its broader role in the envelope stress response, similar to CpxAR (61). Since the commonality of these differing theories centers on zinc, we chose to study ZraR regulatory activity on LB medium supplemented with 1 mM ZnCl$_2$ (57). The ZraR iModulon included only two of the three previously published regulon genes, *zraR* and *zraP*. One of the regulon genes, *zraS*, was not identified by the iModulon. However, ChIP-exo binding peaks were observed upstream of all three regulon genes (*zraSR* and *zraP*), confirming that these genes are under the direct control of ZraR. The ChIP-exo results indicated only one additional direct target gene, *mgtA*, and the remaining 12 binding peaks might be attributed to spurious binding. DEGs were examined to identify indirect targets and expand direct targets (if any) of ZraR. In the presence of 1 mM ZnCl$_2$, 97 genes were found to be differentially expressed between the *zraR* knockout and wild-type strain. Four of these genes (*zraSR*, *zraP*, and *mgtA*) had corresponding upstream binding peaks found by ChIP-exo or were included in the ZraR iModulon. The remaining 93 DEGs can be attributed to indirect ZraR regulation. Therefore, ZraSR seems to be involved in cell migration and carbohydrate metabolism (Fig. 3) similar to CpxAR, further strengthening the cross-regulation of these two TCSs.

Overall, integrating iModulons with ChIP-exo and DEGs enabled us to expand the global regulatory network contribution of each selected TCS and categorize genes in different modes (direct or indirect) of transcription regulation.

**ICA identifies potential novel inducers.** To identify potential novel inducers and interconnection between TCS pathways, we compared the activity levels of each TCS iModulon across all PRECISE experiments. To this end, the iModulon activities of each TCS were analyzed across their expected induction condition(s) in our experiments, as well as the previous conditions of PRECISE experiments. By calculating each TCS iModulon activity for the experiments under control conditions (e.g., LB medium for BaeSR, CpxAR, and ZraSR), a baseline was established for expected TCS behavior in the absence of an induction condition. From there, iModulon activities were calculated for all other experimental conditions across PRECISE and the other TCS experiments of this study, and then these activities were evaluated against the control condition iModulon activity. While many experimental conditions yielded iModulon activities indiscernible from those of the control, several key environmental inducers were shown to have significant differential activity for each TCS (Fig. 4). These results are important for understanding which types of environments are most likely to trigger TCS action. The ability to succinctly examine the activity levels of an iModulon over an ever-growing compendium of conditions enables the broad characterization of the iModulon's response (or lack of response) to each of the conditions. Figure S2 shows the activity profiles for each of the five TCS response regulators (RRs) across all conditions, including PRECISE conditions and the new conditions tested in this study.

When observing the activity levels of ethanol sensors (BaeR and CpxR), we noted that a high average activity level occurred when the wild-type strain was cultured on LB plus 5% ethanol. Interestingly, both the BaeR and CpxR iModulons exhibited increased activity for a previous set of experiments that studied the effect of osmotic stress on *E. coli* K-12 MG1655 in relation to the OmpR regulon (62). Although Bury-Moné et al. (18) had suggested that 0.6 M NaCl did not activate a CpxR and BaeR target gene reporter, our results suggest that osmotic stress (via the addition of 0.3 M NaCl) may induce BaeR and CpxR to some extent. A prior study attempted to identify cross-regulation between CpxAR and OmpR-EnvZ systems (63). In addition to osmotic stress induction, previous PRECISE iron starvation conditions (64) also yielded a significant increase in BaeR iModulon activity, which indicates that BaeR may also be involved in response to iron deficiency (Fig. 4). For CpxR, iModulon activity peaks also occurred across experiments testing for phosphate starvation. The array of conditions inducing the BaeR and CpxR iModulons is reasonable considering the response regulators' extensive functionalities and roles in responding to cell envelope stress (Fig. 4).

KdpE was shown to be directly phosphorylated by PhoR under simultaneous potassium and phosphate limitation (45). Consistent with this, we observed high peak activity of the KdpE iModulon for our TCS experiments conducted on phosphate-depleted M9 medium, in addition to the expected induction of the wild-type strain on TMA plus 0.1 mM KCl (Fig. 4). Another notable iModulon activity increase was observed for a subset of PRECISE experiments that were conducted under osmotic stress (62), which is consistent with KdpE's documented induction under salt stress (42, 43). Compared to KdpE, the PhoB iModulon was active in several of our other TCS experimental conditions, with an especially notable trend of activity increases for all our samples on LB plus 5% ethanol stress. Although PhoB is not generally thought to be a component of the envelope stress response, these iModulon activity increases could potentially be linked to previous research that has found ethanol to induce calcium phosphate crystallization (65, 66) that could limit the amount of freely available phosphate and mirror starvation conditions. For the PRECISE experiments, the PhoB iModulon activity was significantly lower than the PhoB iModulon activity under the phosphate starvation condition in this study. For example, the experiment with the highest PhoB iModulon activity in PRECISE, a *yddM* knockout experiment (67), had a PhoB activity of 2.8, which is only 23% of the phosphate starvation condition PhoB iModulon activity of 11.8. The lower PhoB iModulon activities in the PRECISE experiments is consistent with the absence of phosphate starvation conditions in any of the previous experiments (Fig. 4).

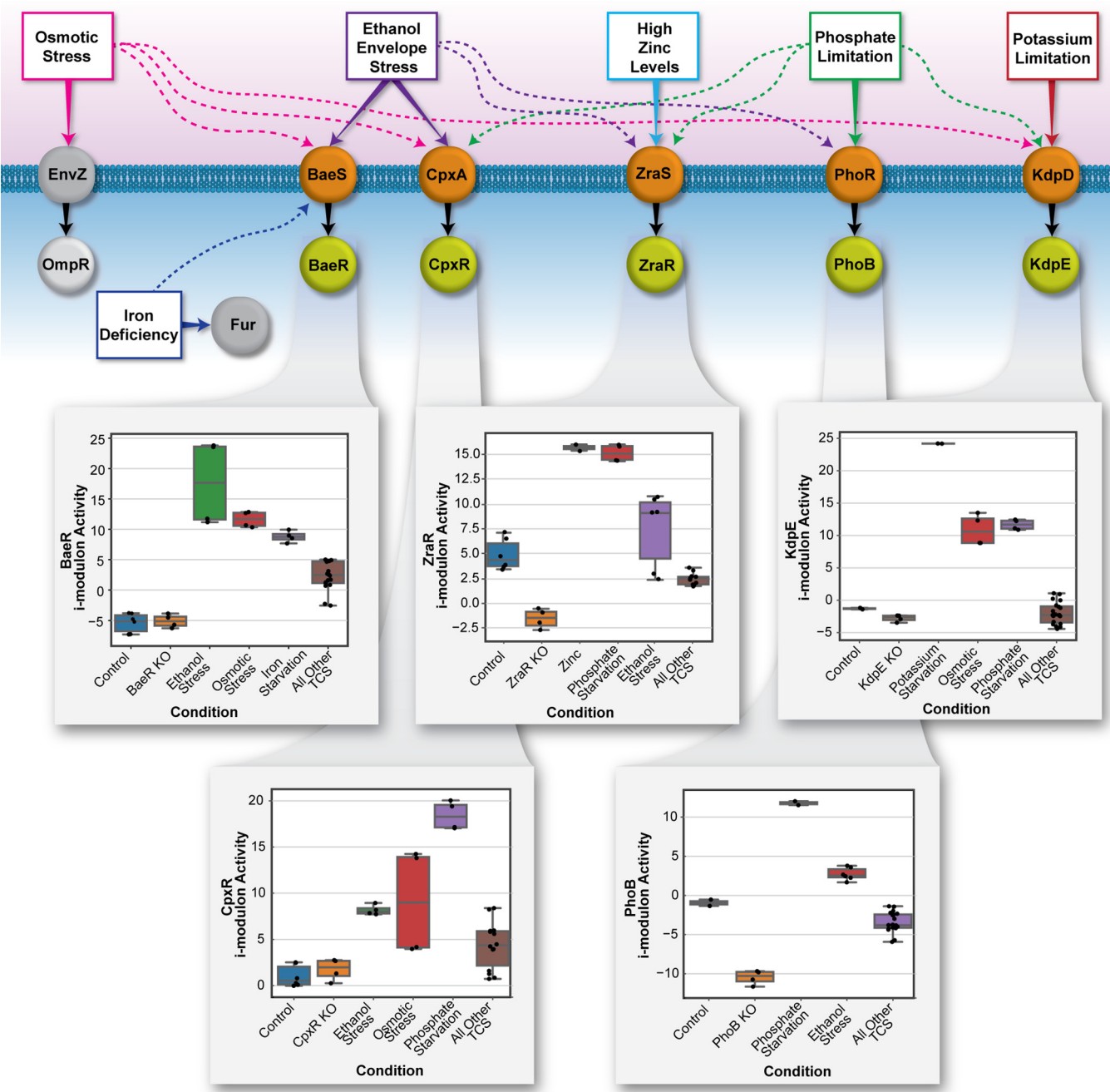

**FIG 4** Cross-regulation among TCS and iModulon activities across selected conditions. The iModulon activity comes from the ICA-derived A matrix and represents the relative strength of the iModulon signal across the compendium of experimental conditions as described by Sastry et al. (7). The boxplots reflect a subset of conditions that are particularly relevant to the given iModulon. The *y* axis represents the iModulon activity level, and the *x* axis represent different experimental conditions. Graphical representations of the full iModulon activity profiles are available in Fig. S2 in the supplemental material.

Notably, the ZraR iModulon includes only the two expected regulon genes, *zraR* and *zraP*, yet this iModulon exhibits significantly heightened activity over a surprisingly wide range of experimental conditions apart from LB plus 1 mM ZnCl$_2$ (Fig. S2). For this study's experiments, the ZraR iModulon activity was high for all experiments conducted in phosphate-depleted M9 medium and most experiments conducted in LB plus 5% ethanol. The high activity of the iModulon across diverse conditions supports the view of ZraR as a broader regulator past simple zinc resistance; specifically, ethanol's activation of the iModulon could be indicative of ZraR's relation to ESR, as previously suggested (61) (Fig. 4).

**TABLE 3** Response regulator induction across various conditions using ICA iModulon activity levels

| Response regulator | ICA-identified induction conditions also found in previous studies | Novel ICA-identified induction conditions |
|---|---|---|
| BaeR | Ethanol stress; osmotic stress | Iron starvation |
| CpxR | Ethanol stress; osmotic stress | Phosphate starvation |
| KdpE | Potassium starvation; osmotic stress | Phosphate starvation |
| PhoB | Phosphate starvation | Ethanol stress |
| ZraR | Zinc; ethanol stress | Phosphate starvation |

Overall, combining our new data set with PRECISE and applying ICA enabled us to identify potential cross-regulation between TCS pathways, which allowed the identification of novel or noncognate TCS-inducer pairs. Both the known induction conditions validated by this study and the novel induction conditions illuminated by this study are summarized in Table 3 below.

**Validation of results.** Before using the various gene data sets to scrutinize the regulatory network of each TCS, the quality of the newly generated data was first evaluated. Namely, the peaks found with ChIP-exo were shown to contain the published consensus motif of each TCS, and the RNA-seq data were analyzed to verify TCS regulon induction under their respective stimulation conditions.

**Verifying ChIP-exo binding peaks.** To validate the binding peaks found by the ChIP-exo analysis, the peaks associated with direct targets from this study were compared to known binding peak motifs using AME (analysis of motif enrichment) from the MEME Suite (68). AME computes position-weight matrix (PWM) score by comparing sequence to the motif. Strong binding sites may have higher PWM scores, while weak binding sites have low PWM scores. The experimentally identified peaks that were found to match the consensus motif as represented by RegulonDB (69) are summarized below in Table 4.

All five of the examined TCSs were shown to have at least one direct target ChIP-exo peak that matches the consensus motif using AME. In accordance with expectations,

**TABLE 4** ChIP-exo peaks for direct target genes that were identified as a match to the consensus motif by AME[a]

| TCS | Peak ID of Match | Operons Associated with Peak | PWM Score | Consensus Motif from RegulonDB (69) |
|---|---|---|---|---|
| BaeR | peak_5 | mdtABCD-baeSR | 3550730 |  |
| | peak_2 | spy | 32 | |
| CpxR | peak_10 | cpxPQ / cpxRA | 12053 |  |
| KdpE | peak_2 | kdpFABC | 34971900 |  |
| PhoB | peak_7 | pstSCAB- phoU | 3910090 |  |
| | peak_0 | phoBR | 6496 | |
| ZraR | peak_9 | zraSR | 2.42e+17 |  |

[a]The position-weight matrix (PWM) score represents the average odds score of a single ChIP-exo peak sequence in comparison to the motif PWM across each position in the sequence.

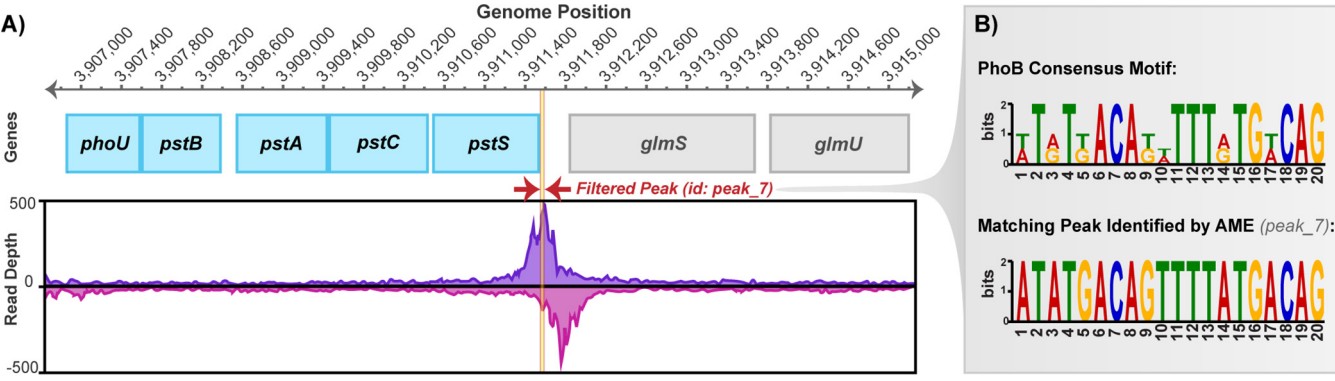

**FIG 5** (A) Example of ChIP-exo peak identification with PhoB peak_7 (*pstSCAB-phoU* operon). (B) Comparison of PhoB binding consensus motif to peak_7, which was identified as a match by AME.

the matching peaks pinpointed by AME included canonical primary targets of each TCS response regulator. A visualization of the peak genomic location and sequence match to the consensus motif is shown below in Fig. 5 for the *pstSCAB-phoU* peak (identifier [id], peak_7) that was identified as a match to the PhoB consensus motif.

**Examining induction with RNA-seq differential gene expression.** The results of this study rely upon the adequate stimulation of the various TCSs under their various induction conditions. As a verification of these induction conditions, a closer look is given to differential gene expression in both the unstimulated and stimulated conditions. For the histidine kinase and response regulator genes of each TCS, the response regulator knockout versus wild-type RNA-seq gene expression is compared under both the unstimulated and stimulated conditions to confirm successful induction (Table 5). A broader comparison of the response regulator knockout versus wild-type expression for the entire direct regulon of each TCS is available in Data Set S5.

Both the histidine kinase and response regulator genes were shown to be differentially expressed at their respective stimulation condition with a log fold change greater than 1.5 and a *P* value less than 0.05 (Table 5). This observation confirms that each of the TCS genes was more expressed in the wild-type strains versus the response regulator knockout (KO) strains under induction. All histidine kinase and response regulator genes other than *cpxR* exhibited a greater magnitude of log fold change than the unstimulated condition. Overall, this serves as an independent validation of successful induction of the histidine kinase elements to the respective stimuli.

## DISCUSSION

*E. coli* uses two-component systems (TCSs) to respond to environmental signals by modulating gene expression. Each TCS employs a signal transduction mechanism to

**TABLE 5** Summary of differential expression of response regulator knockout versus wild-type strains under unstimulated and stimulated conditions for each TCS gene

| TCS | TCS gene | Unstimulated condition, knockout vs wild type | | Stimulated condition, knockout vs wild type | |
|-----|----------|----------------------------|----------|----------------------------|----------|
| | | Log$_2$ fold change | *P* value | Log$_2$ fold change | *P* value |
| BaeR | *baeS* | −0.41 | 2.96E−01 | −4.62 | 4.22E−15 |
| | *baeR* | −8.83 | 6.47E−09 | −9.43 | 4.09E−10 |
| CpxR | *cpxA* | −2.20 | 5.29E−94 | −3.93 | 4.95E−77 |
| | *cpxR* | −12.55 | 3.53E−17 | −11.35 | 2.65E−14 |
| KdpE | *kdpD* | −0.22 | 1.28E−01 | −6.78 | ≅0.00E+00 |
| | *kdpE* | −8.09 | 1.54E−06 | −13.86 | 1.51E−20 |
| PhoB | *phoR* | −1.52 | 1.42E−12 | −3.16 | 1.41E−151 |
| | *phoB* | −9.95 | 1.45E−10 | −15.61 | 8.47E−26 |
| ZraR | *zraS* | −0.02 | 9.45E−01 | −3.18 | 4.00E−139 |
| | *zraR* | −10.67 | 5.38E−13 | −13.33 | 5.44E−19 |

modulate target gene expression when a certain stimulus is received. These signal transduction systems form a part of the global transcriptional regulatory network (TRN). In this study, we identified the TRN contributions of five TCSs in *E. coli*: BaeSR and CpxAR, which were stimulated by ethanol stress, and KdpDE, PhoRB, and ZraSR, which were stimulated by potassium limitation, phosphate limitation, and added zinc, respectively. The recent identification of iModulons in *E. coli*'s transcriptome (7) enables new capabilities to further explore its response to environmental stimuli. Here, this ICA-driven iModulon determination process was implemented in combination with ChIP-exo data analysis to study the TRN contribution of five TCSs. This technique delivered three key results: (i) pinpointing and validation of the target genes for each TCS; (ii) determination of modes of transcriptional regulation of the target genes for each TCS; and (iii) the elucidation of TCS activity across various conditions and novel relationships among TCSs.

The integrative approach of combining ICA, Chip-exo, and RNA-seq data sets provided us with the strong evidence to expand the global regulatory network of each of the five TCSs (BaeSR, CpxAR, KdpDE, PhoRB, and ZraSR) and decipher different modes of transcriptional regulation. With this study, we extend the scope of TRNs beyond direct regulations by defining indirect targets, cross-regulatory networks, and hypothetically functional binding genes. Apart from the classically studied direct targets, genes in cross-regulatory networks and hypothetically functional binding genes fall into new categories that are of particular interest. For example, *ompC* was characterized as a hypothetically functional binding gene target of CpxR, since it was previously reported as part of the CpxR regulon in response to high acetyl phosphate (36) and we identified ChIP-exo binding peaks upstream of it. However, we did not observe any differential expression of this gene or iModulon activity. The lack of response in *ompC* expression could be due to the absence of the high acetyl phosphate condition in our experiments that would be necessary for the change in *ompC* expression to occur. Therefore, we classified this gene in the "hypothetically functional binding gene" category, suggesting a possible regulation of this gene via CpxR despite the lack of substantiation with RNA-seq expression results. Similarly, two genes (*sbcC* and *sbcD*) from the PhoRB network were characterized as hypothetically functional binding genes, which could also indicate that these genes are potential binding targets of PhoB.

The "cross-regulatory network" category included genes that were identified in the iModulon and/or published regulon but could not be confirmed using ChIP-exo methods and/or DEGs. This category of genes is of interest because they show the interwoven and signal-specific nature of the TRN. It could be postulated that cases where the genes were just identified by iModulons and not DEGs were due to a more stringent threshold for calculating DEGs. However, it may be more accurate to consider the unique, albeit complementary, perspectives offered by iModulons and DEGs on the effects of gene knockouts. Specifically, iModulons identify genes that are coexpressed, and likely coregulated, across over 300 expression conditions, whereas DEGs offer a targeted analysis of the effect of a single regulator knockout. Due to the larger data set feeding ICA, patterns can be deciphered that may appear insignificant by DEG analysis alone. For example, we observed five genes in the cross-regulatory network of the CpxR regulon (*alx*, *mdtJ*, *yjfN*, * yliI*, and *yobB*) that were identified by iModulons but not by DEGs because of low *P* value significance (>0.05 threshold). This example highlights the capability of ICA in deciphering the global TRN. When comparing the iModulon and published regulon data sets, it should be noted that the number of iModulon genes is sometimes significantly smaller than the number of regulon genes, as seen with CpxR and PhoB in this study and also many of the other regulators in PRECISE (7). This difference alludes toward the fundamental challenge of determining a global TRN (70) and determining the condition-specific regulon of each regulator. The iModulons may reveal a more specific niche of regulator activity at the subset of conditions available in this study and PRECISE, as opposed to all the many conditions included in RegulonDB. Indeed, it has been shown that both CpxR and PhoB exhibit considerable condition-specific variability in the expression of their regulon genes (71, 72).

In addition to introducing new ways to categorize RNA-seq and ChIP-exo results, we show that applying ICA to transcriptomics data provides a thorough understanding of TCS TRN contributions. Even further, we are able to propose novel relationships between different regulatory elements. For example, we identified that apart from the previously known ethanol stress as an inducer of BaeR, osmotic stress and iron deficiency may also modulate BaeR target gene binding; however, additional experimentation would be necessary to verify this relationship. Other novel relationships were indicated by the increase in CpxR and ZraR iModulon activities under phosphate starvation conditions, which suggests that phosphate-limiting conditions may also act as an inducer for CpxAR and ZraSR two-component systems. Conversely, we also identified a potential involvement of PhoRB and ZraSR in the envelope stress response. One possible explanation of the link between phosphate starvation and envelope stress response could be the role of each of the stimuli in biofilm formation. Specifically, it has been suggested that the phosphate starvation of *E. coli* results in the modification of lipopolysaccharides to increase biofilm formation (73), and CpxAR has been proposed to be induced by biofilm formation due to its sensing of cell adhesion (34). Therefore, the phosphate starvation in this study may have stimulated biofilm formation, which may have in turn induced CpxAR and broader ESR targets. We were also able to confirm previously documented relationships between KdpDE and PhoRB systems (45) wherein the activity of the KdpE iModulon was found to be high under phosphate-limited conditions, in addition to potassium and osmotic stress. Taken together, this study identified several new potential relationships and provided evidence toward suspected relationships by focusing on only five out of the 29 complete TCSs. As the scope increases past these five TCSs, there is potential for identifying additional novel relationships to enhance our knowledge of TRN interconnectedness.

Finally, we validated the gene sets falling in each category by comparing the known consensus motifs using AME. We observed that at least one direct target ChIP-exo peak for each TCS matched the consensus motif using AME. Interestingly, two genes (*sspA* and *sspB*) that were categorized as potentially spurious binding peaks for BaeR were also found to match the consensus BaeR motif. This result could indicate that the *sspAB* operon should be reclassified as a novel binding target of BaeR. The gene *sspA* has been shown to be essential for survival during acid-induced stress (74), so a relationship to the envelope stress-mediating BaeSR TCS does not seem improbable. However, further experimentation would be necessary to validate *sspAB* as a potential binding target of BaeR, especially with the lack of RNA-seq substantiation in this study.

In summary, we suggest that integrating ChIP-exo and DEG analyses with ICA can provide a deeper understanding of the transcriptional regulatory network of *E. coli*. In particular, ICA provides a broad view of TCS activity over diverse conditions, enables the detection of noncognate inducers to response regulators (this study), and can identify redundancy in TF actions (HprSR and CusSR) (our previous study [7]) by showing that different external signals cause expression of the same set of genes. This robust validation of the five TCSs' contributions to the *E. coli* TRN expands the current knowledge base by identifying new potential direct and indirect targets of each TCS RR and by confirming previously predicted or inferred TCS target genes that were found in earlier studies. Looking forward, the horizon of bacterial "awareness" can be examined by incorporating these results into a genome-scale reconstruction of two-component systems which can reliably predict metabolic capabilities under different environmental conditions.

## MATERIALS AND METHODS

**Bacterial strains and growth conditions.** Strains used in this study are *E. coli* K-12 MG1655 and its derivatives (deletion strains and myc-tagged strains). As previously described (75), strains retaining 8-myc were generated by a λ red-mediated site-specific recombination system targeting the C-terminal region of each selected response regulator. Knockout mutants (Δ*baeR*, Δ*cpxR*, Δ*kdpE*, Δ*phoB*, and Δ*zraR*) were generated according to the procedure of Datsenko and Wanner (76), using pKD13 and pKD46 as suggested by the authors (please refer to Table S1 in the supplemental material for a full list of oligonucleotides). All knockouts are complete deletions, except for KdpE; the KdpE locus could potentially produce an 18-bp peptide, since the sequence after the stop codon contains too many A/T-rich

repeats for proper oligonucleotide annealing so the binding site had to be shifted into the coding region (please see Table S1 for the oligonucleotide list). If not stated otherwise, cells were grown in precultures prepared from glycerol stocks overnight and then transferred to fresh medium the next morning. For ethanol sensors (BaeR and CpxR), cells were grown at 37°C in liquid LB medium to an optical density at 600 nm ($OD_{600}$) of 0.5, and then ethanol was added to a final concentration of 5% (wt/vol). To ensure proper binding of RRs to their target genes, cells were grown for 30 min in the presence of ethanol before being collected for ChIP-exo and RNA-seq. This growth condition was chosen according to Bury-Moné et al. (18) (see Table 1 in reference 18) who showed that 5% ethanol led to the strongest induction of $\beta$-galactosidase reporters for CpxR and BaeR among all tested conditions (i.e., 2 mM and 4 mM indole, 3% and 5% ethanol, 0.5 mM dibucaine, 5 mM EDTA, and 0.6 M NaCl). For RNA-seq controls, cells were grown in liquid LB medium and collected at an $OD_{600}$ of 0.5. The KdpDE TCS induces expression of the high-affinity $K^+$ transporter KdpFABC under K-limiting conditions (Schramke et al. [45]). According to Schramke et al. (45), KdpD acts as a phosphatase on KdpE-P and prevents production of the high-affinity $K^+$ transporter at high extracellular $K^+$ concentration (>5 mM). When environmental levels of $K^+$ fall below the threshold for autokinase activation, kdpFABC expression is initiated; however, as long as the intracellular $K^+$ concentration remains high, the KdpD phosphatase activity remains stimulated (Schramke et al. [45]). Therefore, to induce KdpDE, cells were grown in K-sufficient conditions, washed, and then grown under K-limiting conditions to ensure drops in intracellular $K^+$ levels below the threshold level and thus proper induction of the TCS. To do this, cells were grown at 37°C overnight in liquid Tris-maleic acid minimal medium (TMA) (44, 45) supplemented with 115 mM KCl and 0.4% (wt/vol) glucose and then washed twice with TMA containing 0.1 mM KCl and 0.4% (wt/vol) glucose. Cells were inoculated in TMA with 0.1 mM KCl and 0.4% (wt/vol) glucose and collected at an $OD_{600}$ of 0.5 for ChIP-exo and RNA-seq analysis. For RNA-seq controls, cells were grown in liquid TMA supplemented with 115 mM KCl and 0.4% (wt/vol) glucose and collected at an $OD_{600}$ of 0.5. For induction of the PhoRB TCS, we chose to use M9 minimal medium without any phosphate to ensure proper induction of the system. To our knowledge, this condition was not used before. To induce phosphate-limiting conditions, cells were grown in liquid M9 minimal medium (containing phosphate) until an $OD_{600}$ of 0.5 was reached. Growing cells until the mid-exponential growth phase was necessary, because the lack of phosphate led to growth arrest. The cells were then washed three times with M9 minimal medium without phosphate (M9-P) (without $Na_2HPO_4$ and $KH_2PO_4$) and incubated in M9-P for 60 min (about the doubling time of strain MG1655 in M9 minimal medium) at 37°C. Cells were then collected for ChIP-exo and RNA-seq analysis. For RNA-seq controls, ΔphoB cells were grown in liquid M9 minimal medium and collected at an $OD_{600}$ of 0.5 (WT samples were taken from reference 7; see Table 1). For ZraSR, cells were grown at 37°C in liquid LB medium containing 1 mM $ZnCl_2$ by the method of Leonhartsberger et al. (57) but without the addition of glucose to the LB medium. At an $OD_{600}$ of 0.5, cells were collected for ChIP-exo and RNA-seq analysis. For RNA-seq controls, cells were grown in liquid LB medium and collected at an $OD_{600}$ of 0.5. Samples for ChIP-exo and RNA-seq were taken independently.

**RNA-seq.** For RNA-seq, 3 ml of culture was mixed with 6 ml of RNAprotect bacterial reagent (Qiagen) and processed according to the manufacturer's instructions. Cell pellets were frozen and stored at −80°C until processed. RNA was extracted using the Zymo Research Quick RNA fungal/bacterial microprep kit, according to the manufacturer's instructions. rRNA was removed from total RNA preparations using RNase H. First, traces of genomic DNA were removed with a DNase I treatment. Then, secondary structures in the rRNA were removed by heating to 90°C for 1 s. A set of 32-mer DNA oligonucleotide probes complementary to 5S, 16S, and 23S rRNA subunits and spaced nine bases apart were then annealed at 65°C followed by digestion with Hybridase (Lucigen), a thermostable RNase H. Hybridase was added at 65°C, the reaction mixture was incubated for 20 min at that temperature, then heated again to 90°C for 1 s to remove remaining secondary structures, and finally returned to 65°C for 10 min. The reaction was quickly quenched by the addition of guanidine thiocyanate while still at 65°C before purifying the mRNA with a Zymo Research RNA Clean and Concentrator kit using their 200-nucleotide (nt) cutoff protocol. Carryover oligonucleotides were removed with a DNase I digestion which was started at room temperature and gradually increased to 42°C over a half hour. This was followed up with another column purification as stated above.

Paired-end library preparation was done using the KAPA RNA HyperPrep kit following the manufacturer's instructions with an average insert size of 300 bp. The libraries were then analyzed on an Agilent Bioanalyzer DNA 1000 chip (Agilent). RNA-seq libraries were sequenced on the NextSeq 500 (Illumina) at the Salk Institute for Biological Studies, La Jolla, CA, USA. RNA-seq experiments were performed in duplicates.

**ChIP-exo experiments.** To identify binding sites for each response regulator under their respective induction conditions (see above and Table 1), ChIP-exo experiments were performed as described previously (64). Specifically, cells were cross-linked in 1% formaldehyde and DNA bound to each response regulator was isolated by chromatin immunoprecipitation (ChIP) with the specific antibodies that recognize the myc tag (9E10; Santa Cruz Biotechnology) and Dynabeads Pan Mouse IgG magnetic beads (Invitrogen). This was followed by stringent washing as described earlier (77). To perform on-bead enzymatic reactions of the ChIP-exo experiments, ChIP materials (chromatin-beads) were used (5). The method has been described in detail previously (64, 67). ChIP-exo experiments were performed in duplicates. ChIP-exo libraries were sequenced on the NextSeq 500 (Illumina) at the Sanford Burnham Prebys Medical Discovery Institute, La Jolla, CA, USA, or on the NextSeq 500 (Illumina) at the Salk Institute for Biological Studies.

**ChIP-exo processing and peak calling.** The ChIP-exo sequence reads were mapped onto the reference genome (E. coli MG1655 GenBank accession no. NC_000913.3) using bowtie (78) with default

options as also described previously (64, 67). This generated SAM (sequence alignment/map) output files. Peak calling from biological duplicates for each experimental condition was done using the MACE program (79). Two-level filtering of ChIP-exo data was done: first, we removed any peaks with a signal-to-noise (S/N) ratio of less than 1.5 to reduce false-positive peaks. The top 5% of the signals at genomic positions was set as the noise level as previously described (64, 67). MetaScope (https://sites .google.com/view/systemskimlab/software) was used to visualize the peaks. To further reduce the false-positive peaks, a cutoff analysis was done for each regulator on the S/N ratio (see Fig. S1 in the supplemental material). We removed any peaks that had a S/N ratio below the threshold. Thereafter, the peak was assigned to the nearest operon on both strands. Operons located within 500 bp of the peak were considered. Specifically, if a peak was upstream of an operon, we considered all the genes in that operon to be regulated by that response regulator.

**Analysis of motif enrichment for consensus in ChIP-exo peaks.** After the final S/N thresholding and 500-bp cutoffs were applied to filter the ChIP-exo data, the resulting binding peaks were buffered with a margin of 20 bp on either end of their sequence and then compared to the published consensus motif to identify matching sequences (Table 4). The search was performed by analysis of motif enrichment (AME) from the MEME suite (68) using the options "--scoring avg --method fisher --hit-lo-fraction 0.25 --evalue-report-threshold 10.0 --control --shuffle-- --kmer 2." The "--scoring avg" option specifies that the method of scoring a sequence compared to the motif's PWN consists of averaging the motif odds score across all positions in the sequence. The "--method fisher" option selects the one-tailed Fisher's exact test method to evaluate motif enrichment. The fraction of maximum log odds for scoring a potential match was specified as 0.25 using "--hit-lo-fraction 0.25," and the E value for reporting was arbitrarily set at 10 (no TCS had more than 10 identified sequences) using "-evalue-report-threshold 10.0." A zero-order Markov sequence model was created by AME to normalize for biased distributions; additionally, AME created a control sequence by shuffling the letters in each of the input sequences while preserving 2-mers using "--control --shuffle-- --kmer 2."

**Differential gene expression.** RNA-seq sequence reads were mapped onto the reference genome (*E. coli* MG1655 GenBank accession no. NC_000913.3) using bowtie 1.1.2 (78) with the following options "-X 1000 -n 2 − 3 3," where X is maximum insert size, n is number of mismatches, and −3 3 denotes trimming of three base pairs at the 3′ end. Read count was performed using *summarizeOverlaps* from the R *GenomicAlignments* package, using options "mode="IntersectionStrict," singleEnd=FALSE, ignore.strand=FALSE, preprocess.reads=invertStrand" (80). Differential gene expression was identified by *DESeq2* (81). Genes with $\log_2$ fold change of >1.5 and a false discovery rate (FDR) value of <0.05 were considered differentially expressed genes. Genes with *P* values assigned "NA" (for not available) based on extreme count outlier detection were not considered potential DEGs.

**Independent component analysis.** To identify independent sources of gene expression control, independent component analysis (ICA) was applied to the combined data set that consisted of all 278 RNA-seq gene expression data from PRECISE (7) and 32 data sets from this study which included gene expression data of knockout mutants and wild-type strains under stimulated and unstimulated conditions. Genes with less than 10 fragments mapped per million reads across the entire data set were removed from the RNA-seq data sets to reduce noise. FastICA was performed 100 times with random seeds and a convergence tolerance of $10^{-6}$. For each iteration, we constrained the number of components to the number that generates 99% variance in principal-component analysis (PCA). We further used DBSCAN to cluster the resulting components to identify a set of robust independent components using a minimum cluster seed size of 50. We repeated this process five times, and only the components that occurred consistently in each run were taken. iModulons were extracted as previously described (7).

**iModulon characterization.** We checked for the significant genes in each iModulon and mapped it to the set of genes regulated by a specific regulator or transcription factor. This comparison was done using one-sided Fisher's exact test (FDR $< 10^{-5}$). The regulon/regulator with the lowest *P* value was given the name of the iModulon. The regulon/regulators that were not characterized by this method were further analyzed through Gene Ontology (GO) enrichment. In this case, significant genes in each iModulon were compared to genes in each GO term, and a *P* value was assigned using the one-sided Fisher's exact test (FDR $< 10^{-5}$). The GO term with the lowest *P* value was assigned as the name of the iModulon.

**Data availability.** The ChIP-exo and RNA-seq data sets have been deposited into GEO with the accession number GSE143856. All other data are available in the supplemental material.

## SUPPLEMENTAL MATERIAL

Supplemental material is available online only.

**FIG S1**, PDF file, 0.4 MB.
**FIG S2**, PDF file, 0.3 MB.
**TABLE S1**, PDF file, 0.04 MB.
**DATA SET S1,** XLSX file, 0.1 MB.
**DATA SET S2,** XLSX file, 0.03 MB.
**DATA SET S3,** XLSX file, 0.2 MB.
**DATA SET S4,** XLSX file, 0.05 MB.
**DATA SET S5,** XLSX file, 0.04 MB.

## ACKNOWLEDGMENTS

We thank Irina Rodionova for helpful discussions. We thank Marc Abrams for reviewing the manuscript and providing constructive suggestions.

This work was funded by the Novo Nordisk Foundation grant NNF10CC1016517.

K.S.C. and B.O.P. designed the study. J.A.K. designed the strains, generated the strains, and performed the experiments. Y.G. and R.S. assisted in generating RNA-seq and ChIP-exo data. K.S.C. and K.D. processed the RNA-seq and ChIP-exo data and performed computational analysis. A.V.S. and Y.S. contributed to data analysis. K.S.C., K.D., A.V.S., and B.O.P. wrote the manuscript, with contributions from all other authors.

We declare that we have no financial competing interests.

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
