## [Reviewer comments · mSystems]

Elucidation of regulatory modes for five two-component systems in *Escherichia coli* reveals novel relationships

Kumari Sonal Choudhary, Julia Kleinmanns, Katherine Decker, Anand Sastry, Ye Gao, Richard Szubin, Yara Seif, and Bernhard Palsson

Corresponding Author(s): Bernhard Palsson, University of California, San Diego

Review Timeline:

Submission Date:

September 24, 2020

Accepted:

October 20, 2020

Editor: Rafael Silva-Rocha

Reviewer(s): The reviewers have opted to remain anonymous.

Transaction Report:

DOI: <https://doi.org/10.1128/mSystems.00980-20>

Point by Point Response to Reviewer comments:

Please refer to the point by point Authors response below.

Reviewer #2 :

In this paper, the authors used a combination of experimental work (RNAseq and ChIP-exo) and re-analysis of previously obtained dataset using independent component analysis (ICA) to assess the regulons of 5 TCS, and their interconnection in E.coli. This kind of multiple-approach analyses could be of interest for the study of the transcriptional landscape of other species, especially the ones for which huge dataset are available. Here, integration of the different types of results and analyses provide interesting information on the studied TCS. It allows the authors to better categorize the genes previously found to belong to a regulon into more specific categories depending on whether they appeared in one or more of the analyses. Notably, by comparing the results obtained by ICA, RNAseq and ChIP-exo, they distinguish between direct and indirect targets of each particular TCS. The authors also provide some evidence that these 5 different TCS might be more interconnected than was previously thought. The paper is nicely written, and the method section is clear and well detailed. I have only a few comments. The major ones should be addressed by the authors.

Major points:

- I would have liked the authors to comment, for example in the discussion, on the relatively low level of overlap between the regulons obtained with each different methods. It asks the question of what is the "real" composition of the regulon of each considered TCS.

Author's response: We have added the explanation in the discussion, line numbers 501-509, Page numbers 13 of the Marked-Up manuscript (without figures), to discuss the lower overlap between genes in iModulons and the published regulons. Specifically, the difference is observed because regulonDB has TRNs spanning multiple conditions, in comparison to condition specific activity reported via iModulons.

- 1482-486: "For the respective unstimulated conditions, some of the TCS genes were also differentially expressed, including all the response regulator genes and the cpxA and phoR histidine kinase genes. This result shows that all the response regulator genes and two of the histidine kinase genes were being expressed in the wild type even without induction."
 - As the KO strains are proper deletion of the RR genes, it is normal (and expected) to have no reads from the RNAseq matching in the RR genes and therefore to obtain a big negative fold change for this genes. This do not allow for any conclusions to be made on the expression of these genes in the WT. In the WT, some transcriptional error (random promoter activity, failure of transcription to stop at an upstream terminator...) could

lead to the production of a few mRNAs, or even pieces of them, from the RR genes and lead to a few reads via RNAseq (even surpassing the authors' threshold of 10 reads per million). That would not be a true expression of the RR genes in the WT.

Author's response: We thank the reviewer for pointing this out. We have corrected the sentences as suggested. We removed the sentences which draw conclusions on the expression of TCS genes in unstimulated conditions. Specifically, in this paper we have used the table (Table 5) to compare the gene expression under stimulated and unstimulated conditions.

1513-520: "ompC was characterized as a hypothetically functional binding gene target of CpxR, since it was previously reported as part of the CpxR regulon in response to high acetyl phosphate (36) and we identified ChIP-exo binding peaks upstream of it. However, we did not observe any differential expression of this gene or iModulon activity."

- The authors hypothesize that this might mean that ompC might be repressed when CpxR is activated in response to ethanol. But that doesn't explain why ompC was not detected in iModulon, which combine all the tested conditions of the PRECISE dataset. The simplest explanation of the non-detection of ompC as regulated via CpxR in their RNAseq and iModulon, despite the ChIP-exo results and the previous detection under acetyl phosphate induction, would be that the regulation of ompC via CpxR occurs specifically in conditions not comprised in this study and the PRECISE dataset (for example, high acetyl phosphate would be one of those conditions).

Author's response: We thank the reviewer for pointing this out. We have corrected the sentence to suggest that change in *ompC* expression may require high acetyl phosphate conditions, line number 480-482 Page 13 of the Marked-Up manuscript (without figures).

Minor points:

- fig 2 and fig 3: it would be nice for the reader if the name of the "direct target" genes were on the figure as well as in the text.

Author's response: We have updated both the figures as suggested by the reviewer.

- 139: change to plural: E. coli's two-component systems (TCSs) modulate target gene expression

Author's response: This has been revised

- I101-102: "we aimed to characterize global TRN contributions of these five TCSs (BaeSR, CpxAR, KdpDE, PhoRB, and ZraSR) in E. coli MG1655" > "these" would indicate that the TCS in question were just mentioned, but they are not. So, I would rephrase as: we aimed to characterize global TRN contributions of five TCSs in E. coli MG1655: BaeSR, CpxAR, KdpDE, PhoRB, and ZraSR.

Author's response: Thank you for your suggestion. We have corrected this.

- I108: cite Figure 1B

Author's response: Revised as suggested.

- I131: "Table 1: List of strains used in this study" > Table 1: List of strains and culture conditions used in this study

Author's response: Revised as suggested.

- I201-203: "In most of the cases we did identify genes being regulated by BaeR knockout but they either did not pass the P-value 202 threshold (< 0.05) or log 2 fold change threshold (1.5)." > If the p value and/or the fold change do not pass the threshold then the genes are not regulated. Something cannot be "almost significant". One explanation for the lack of regulation on a gene that was expected to be regulated might simply be a lack of stringency of the inducing condition, or a too short induction phase.

Author's response: We thank the reviewer for pointing out. We have revised accordingly in Page 6, lines 197-199 of the Marked-Up manuscript (without figures).

- I208: I would use "shared genes" instead of "common genes" as "common" also means usual or ordinary it might be a bit confusing.

Author's response: We have corrected this as per reviewer's suggestion.

- I294-296: "DEGs further added three extra genes (adeP, uxaA, and uxaC) to the direct regulatory network of KdpE, amounting to six genes under direct regulation of KdpE." > if I understood correctly, those are in the "direct regulation" category because they appear both by ChIP and RNAseq, so this should be stated in this sentence. For example: Comparing ChIP-exo and DEGs results further added...

Author's response: We thank the reviewer for this suggestion. We have made this change.

- I477: in table 5, in the kdpD line, the last column has a weird font

Author's response: We have revised the text as suggested.

- I555: adhesions

Author's response: We have changed it to singular as suggested.

- I701-704: the font is different on those lines

Author's response: We have revised the text as suggested.

Reviewer #3 (Comments for the Author):

The work by Choudhary et al used data integration to investigate two-component systems in *E. coli*. This is a very interesting work and of general interest, but the key component of the analysis lack statistical support as commented below. The authors should clarify these points before making the major claims of this work.

Major

- The key point of the manuscript is the RNASeq and ChIP-exo analysis. But how many samples were analyzed by RNASeq? It says duplicates (only 2 biological samples per condition?). How can the author perform a reliable statistical analysis with this? The main text cites p-val and fold change but this was not mentioned in the text. The same thing is applied to ChIP-exo.

Author's response: Although it is generally common to use three replicates for statistical analysis, two replicates is the new standard in microbial omics. Our lab has rigorously tested the impact of sample size on microbial omics data and have conclusively shown that two-biological replicates have extremely high confidence to derive statistical analysis like p-values and log2 fold change, especially when low-variation is observed between two replicates for the same condition. We have extremely high correlations between samples ($R^2 = 0.95$). Although it is true that fewer replicates will lead to lower sensitivity, this has not limited any of our prior studies as we can still perform differential expression analysis using standard pipelines (i.e. DESeq2). Some of the selected peer-reviewed published articles which have used duplicate and derived statistics are below for your reference:

<https://www.nature.com/articles/ncomms5910>

<https://www.nature.com/articles/ncomms8970>

<https://academic.oup.com/nar/article/46/20/10682/5078243>

For Chip-exo data, our aim was to get the qualitative result to check if there was a binding site or not, rather than quantitative measurement of how strong the binding was.

Hence, we justify that it is appropriate to use two-biological replicates in cases where high variance is not observed between similar conditions and statistics derived from these analyses are hence valid.

Minor

- Figure 1. This figure is very nice, but the font size is too small, which makes it difficult to read. I recommend increasing font size and perhaps reducing the amount of words in Fig. 1A.

Author's response: We have increased the font size as suggested. The increase in font size did not require compromise on reduction on the amount of words in the figure for clarity.

- Lines 150-151. What is a TFs hot spot? Any reference for that? Could you develop it a little better?

Author's response: The word hot spot is used to indicate regions that are bound by several transcription factors and does not imply the hotspots found in mammalian systems.

Figure 4. The figure legend is not self explanatory. What is iModulon activity? Please enhance the legend.

Author's response: We have expanded the legend as per reviewer's suggestion.

- Lines 414 to 415. What is a iModulon negative or minimally positive? Please enhance the description of these results and if possible a more quantitative way.

Author's response: We have elaborated the description of this result on Page 10-11, line numbers 390-396 of the Marked-Up manuscript (without figures).

Table 4. How was the PWM score calculated? The numbers are drastically different. Additionally, adding how many sequences were used to construct the logo would be relevant. Also check the logos, since many positions seem to be conserved in 100% of the sequences but are not reaching 2 bits of score.

Author's response: We have added the explanation of how AME calculates score and how it is relevant in understanding sequence motifs on Page 11, Line numbers 424-426 of the Marked-Up manuscript (without figures). We have also fixed the logos to go upto 2 bits in Table 4.

- Table 5 and line 482. What is RR knockout? Please define. Also, there are several different ways to refer to the mutant strains through the text (deleted strains, knockout, KO, etc.). Please standardize according to microbiological standards.

Author's response: We have revised this as suggested.

- Lines 689 to 697. Please add the parameters used for AME analysis as you did for bowtie.

Author's response: We have added the parameters for AME analysis on Page 18, line numbers: 678-688, of the Marked-Up manuscript (without figures).

- The supplementary material is a little chaotic and should be condensed and better presented. Supp Figures are not well presented and this part of the manuscript should be enhanced too.

Author's response: We have combined Dataset S4-S7 into one supplementary dataset S4 and have added the legends to the Supplementary Figures 1 and 2. We hope this enhances the clarity and is more readable.

October 19, 2020

Dr. Bernhard O Palsson
University of California, San Diego
Bioengineering
San Diego, CA 92093

Re: mSystems00980-20 (Elucidation of regulatory modes for five two-component systems in *Escherichia coli* reveals novel relationships)

Dear Dr. Bernhard O Palsson:

Your manuscript has been accepted, and I am forwarding it to the ASM Journals Department for publication. For your reference, ASM Journals' address is given below. Before it can be scheduled for publication, your manuscript will be checked by the mSystems senior production editor, Ellie Ghatineh, to make sure that all elements meet the technical requirements for publication. She will contact you if anything needs to be revised before copyediting and production can begin. Otherwise, you will be notified when your proofs are ready to be viewed.

Sincerely,

Rafael Silva-Rocha
Editor, mSystems

Journals Department
Supplemental Dataset S4: Accept
Supplemental Table S1: Accept
Supplemental Figure S2: Accept
Supplemental Dataset S2: Accept
Supplemental Figure S1: Accept
Supplemental Dataset S3: Accept
Supplemental Dataset S1: Accept
Supplemental Dataset S5: Accept